# p-Coumaric Acid Has Protective Effects against Mutant Copper–Zinc Superoxide Dismutase 1 via the Activation of Autophagy in N2a Cells

**DOI:** 10.3390/ijms20122942

**Published:** 2019-06-16

**Authors:** Tomoyuki Ueda, Taisei Ito, Hisaka Kurita, Masatoshi Inden, Isao Hozumi

**Affiliations:** Laboratory of Medical Therapeutics and Molecular Therapeutics, Gifu Pharmaceutical University, 1-25-4 Daigaku-nishi, Gifu 501-1196, Japan; 126014@gifu-pu.ac.jp (T.U.); 156004@gifu-pu.ac.jp (T.I.); kurita@gifu-pu.ac.jp (H.K.); inden@gifu-pu.ac.jp (M.I.)

**Keywords:** p-coumaric acid, amyotrophic lateral sclerosis, oxidative stress, endoplasmic reticulum stress, copper–zinc superoxide dismutase 1, autophagy

## Abstract

Amyotrophic lateral sclerosis (ALS) is a neurodegenerative disease characterized by the selective death of motor neurons. In previous our study, an ethanol extract of Brazilian green propolis (EBGP) prevented mutant copper–zinc superoxide dismutase 1 (SOD1^mut^)-induced neurotoxicity. This paper aims to reveal the effects of p-coumaric acid (p-CA), an active ingredient contained in EBGP, against SOD1^mut^-induced neurotoxicity. We found that p-CA reduced the accumulation of SOD1^mut^ subcellular aggregation and prevented SOD1^mut^-associated neurotoxicity. Moreover, p-CA attenuated SOD1^mut^-induced oxidative stress and endoplasmic reticulum stress, which are significant features in ALS pathology. To examine the mechanism of neuroprotective effects, we focused on autophagy, and we found that p-CA induced autophagy. Additionally, the neuroprotective effects of p-CA were inhibited by chloroquine, an autophagy inhibiter. Therefore, these results obtained in this paper suggest that p-CA prevents SOD1^mut^-induced neurotoxicity through the activation of autophagy and provides a potential therapeutic approach for ALS.

## 1. Introduction

Amyotrophic lateral sclerosis (ALS) is a rapid, progressive neurodegenerative disease that is characterized by muscle weakness, paralysis, and respiratory failure, leading to death within 3–5 years. While about 90%–95% of ALS is sporadic (sALS), about 5%–10% is familial (fALS). fALS is identified by mutations in several genes such as *SOD1*, *C9ORF72*, *TARDBP*, and *FUS*. In Europe, most fALS mutations are in *C9ORF72* (33.7%), *SOD1* (14.8%), *TARDBP* (4.2%), and *FUS* (2.8%). In contrast, Asian fALS mutations are in *SOD1* (30%), *FUS* (6.4%), *C9ORF72* (2.3%), and *TARDBP* (1.5%) [1,2]. The causes of ALS are unknown; however, misfolded proteins are abnormally accumulated in the cytoplasm of motor neurons in ALS patients when examined from a clinical viewpoint. As a potential reason for this, dysfunctional protein degradation may be related to the onset and progression of ALS [3]. The ubiquitin proteasome system (UPS) and autophagy pathway involved in mutant copper–zinc superoxide dismutase 1 proteins (SOD1^mut^) form insoluble aggregations in motor neurons [4,5]. It remains unclear how SOD1^mut^ aggregation causes motor neuron death. It has been reported that the subcellular accumulation of excess SOD1^mut^ impairs the degradation ability of the UPS [6,7]. In motor neurons of ALS patients, typical adverse effects such as mitochondrial dysfunction, oxidative stress, and endoplasmic reticulum (ER) stress are evoked by the dysfunction of the UPS with excess aggregations [5,8]. SOD1^mut^ aggregates accumulate in organelles such as the mitochondria and ER [8,9]. Accumulation of SOD1^mut^ in these organelles induces oxidative stress and ER stress, and then oxidative stress and ER stress cause a further increase in the formation of insoluble aggregates of SOD1^mut^ [10,11]. Several studies have shown that this excessive oxidative stress leading to neuronal cell death is caused by the accumulation of misfolded SOD1 [12,13]. In addition, it has been reported that the activation of autophagy suppresses motor neuronal cell death through the clearance of SOD1^mut^ aggregations in cellular and mouse models of ALS [14,15]. Therefore, the activation of autophagy may represent a potential therapeutic approach for ALS.

Propolis is a resinous substance, and it is made from the tree bud exudates and sap of various botanic plants and the secretion of honeybees. There are many kinds of propolis, classified by smell and color (green, yellow, red, black, and brown), which depend on the vegetable source, season, and country of origin. Several studies have demonstrated numerous pharmacological properties and biological activities of propolis, such as antibacterial, anti-inflammatory, and antioxidative effects [16,17,18,19,20]. A reason for these properties is that many components such as flavonoids and cinnamic acid derivatives are present in propolis [21,22]. In addition, our previous study showed that an ethanol extract of Brazilian green propolis (EBGP) and kaempferol contributed to the clearance of SOD1^mut^ aggregations via the activation of the autophagic pathway and that EBGP and kaempferol have a neuroprotective effect against SOD1^mut^-induced neurotoxicity [23]. However, the effect of other active ingredients of EBGP against SOD1^mut^-related toxicity has not yet been investigated.

Among the active ingredients of EBGP, p-coumaric acid (p-CA), a phenolic class compound, is also widely included in vegetation and many human foods. Many researchers have explained the versatile medicinal activities of p-CA, including antioxidant, cardioprotective, antimelanogenic, antimutagenic, antiplatelet, anti-inflammatory, and immunomodulatory actions [24,25,26,27,28,29,30]. A recent study showed that p-CA induces autophagy activation, although the mechanism remains unclear [31,32]. Therefore, because autophagy is well known as a characteristic event in ALS, we chose p-CA among these active components in EBGP and examined the effects of p-CA against SOD1^mut^-related toxicity from the viewpoint of autophagy.

## 2. Results

### 2.1. p-CA Reduced Cytoplasmic Aggregation of SOD1^mut^ and Protected against SOD1^mut^-Associated Neurotoxicity

Currently, over 160 types of SOD1 pathogenic mutations have been identified in ALS patients [33,34]. Among those, pathogenic SOD1^G85R^ has been studied frequently [34,35]. Based on our previous studies, we know that SOD1^G85R^-transfected Neuro2a cells (N2a cells) form subcellular aggregates and have neurotoxicity [23,36]. To examine the protective effect of p-CA against SOD1^G85R^ aggregates, we automatically counted the number of SOD1^G85R^ aggregates using an IN Cell Analyzer 2200. p-CA was found to significantly decrease these SOD1^G85R^ aggregates (Figure 1A,B). In addition, separation of the protein fraction using Triton X-100 was performed according to the method in our previous studies [23,36]. Western blot analysis showed that the quantity of 1% Triton X-100-insoluble SOD1 aggregates was reduced by p-CA (Figure 1C,D). Next, to investigate the effect of p-CA against SOD1^G85R^-induced neurotoxicity, we investigated the cell viability by 3-(4,5-dimethylthiazol-2-yl)-2,5-diphenyl-2H-tetrazolium bromide assay (MTT assay). The cell viability increased in the 100 nM and 1 µM, but not 1 nM, p-CA-treated groups as compared with the non-treated group (Figure 1E). These results were further supported by cell toxicity assay. SOD1^G85R^ induced an increase of cell toxicity, while p-CA treatment effectively attenuated the neurotoxicity, similar to the result of the MTT assay (Figure 1F). From these results, p-CA has a neuroprotective effect against SOD1^G85R^-associated neurotoxicity.

### 2.2. p-CA Attenuated SOD1^G85R^-Associated Oxidative and Endoplasmic Reticulum (ER) Stress

It was previously reported that p-CA has antioxidative capacity [30]. As excess oxidative stress is present in the motor neurons of ALS patients, the amount of ROS was detected in SOD1^G85R^ cells treated with p-CA using CellROX Green, a fluorescent probe used to investigate oxidative stress, and MitoSOX Red, a fluorescent probe used to measure superoxide in mitochondria [37]. The increase in the CellROX Green fluorescence level in transfected SOD1^G85R^ cells was significantly decreased by p-CA treatment (Figure 2A,B). p-CA treatment also obviously decreased the level of MitoSOX Red fluorescence (Figure 2C,D). To reveal that p-CA directly scavenged ROS, we measured the hydroxy radical spin by ESR assay. p-CA significantly attenuated the signal intensity of hydroxy radicals (Figure 2E,F).

ER stress is an exacerbating mechanism of ALS and has been proposed as a major pathological reaction in various experimental models of the disease [38]. To examine the effect of p-CA against SOD1^G85R^-induced ER stress, Western blot analysis was performed with immunoglobulin heavy chain-binding protein (BiP) and transcription factor C/EBP homologous protein (CHOP) antibodies. Treatment with p-CA significantly inhibited the induction of the ER stress markers, BiP and CHOP, by SOD1^G85R^ (Figure 3A,B,C).

### 2.3. p-CA Exerted a Neuroprotective Effect against SOD1^G85R^ Aggregates via the Activation of Autophagy

To analyze the mechanism of the neuroprotective effect of p-CA against SOD1^G85R^-associated neurotoxicity in this experiment, we focused on autophagy, one of the protein degradation pathways. Previous studies have reported that p-CA induces autophagy [31,32]. Therefore, we performed a neurochemical analysis using the LC3 antibody, which is the most famous autophagy marker. Western blot analysis showed that p-CA increased the formation of LC3-II (Figure 4A,B). We also examined the protein level of p62, another selective marker of autophagy. Western blot analysis showed that p-CA decreased the protein level of p62 (Figure 4A,B). In addition, p-CA increased the level of LC3-II and decreased the protein level of p62 with N2a cells transfected with SOD1^mut^ (Figure 4C,D). Chloroquine (CQ), an autophagy inhibitor, prevented the reduction of cytoplasmic aggregation of SOD1^G85R^ caused by p-CA (Figure 4E,F). To investigate whether the neuroprotective effects of p-CA are associated with autophagy, we performed the MTT assay and LDH release assay in the presence of CQ. The protective effect of p-CA was significantly prevented by CQ treatment (Figure 4G,H). From these results, p-CA reduced the quantity of subcellular aggregates through the upregulation of autophagy, which then prevented SOD1^G85R^-associated neurotoxicity.

## 3. Discussion

Several studies have been reported the usefulness of p-CA to human health and the effects of p-CA, such as its anti-inflammation, anticancer, and antioxidant properties [24,29,32]. However, little study on p-CA in association with neurodegenerative disorders such as ALS has been reported. Dominant mutations in *SOD1* are frequently the cause of the inherited form of ALS [1]. Although the pathological mechanism of ALS is unclear, misfolded SOD1 accumulates in the motor neurons of both sALS and fALS patients [39,40]. It is quite possible that reducing the number of misfolded SOD1 can prevent the onset and progression of ALS [41]. This study aimed to investigate the relation between p-CA and SOD1^G85R^-associated neurotoxicity in a cellular model of ALS.

In this study, consistent with previous reports, we showed that p-CA induced autophagy. In general, AMP-activated protein kinase (AMPK) inhibits the activation of the mammalian target of rapamycin (mTOR), which negatively regulates autophagy. In our previous study, we showed that the kaempferol contained in EBGP potentially induced autophagy via AMPK phosphorylation [23]. Other reports have shown that p-CA activates AMPK [42,43]. From these reports, p-CA may induce autophagy via the AMPK–mTOR pathway. In addition, we previously found that kaempferol did not activate autophagy via the Protein Kinase B (AKT)–mTOR pathway [23]. The AKT signal induced mTOR activity and then inhibited autophagy [44]. In our study, unfortunately, we did not examine whether p-CA inhibited the AKT signal. However, p-CA is known to inhibit the AKT signal [45]. Thus, we can infer that p-CA may induce autophagy via not only the AMPK–mTOR pathway but also the AKT–mTOR pathway.

AMPK is inactivated in models of SOD1^mut^ [46,47,48]. Moreover, cystatin C restores the inactivation of AMPK to control levels and has neuroprotective effects against SOD1^mut^-associated neurotoxicity [47,48]. From these reports, we suggest that the activation of AMPK plays a very important role in preventing the progression of ALS. Therefore, we propose that p-CA is a very useful compound in the search for new drugs.

Autophagy contributes to reducing the quantity of SOD1^mut^ aggregates [23,49]. Preventing degenerated protein accumulation represents an important step to achieving neuroprotection. In addition, reducing the occurrence of misfolded proteins, including SOD1^mut^, is also important, and whether p-CA activates the UPS will need to be investigated in the future [50,51]. If p-CA can activate the UPS, p-CA would then reduce the amount of SOD1^mut^ intracellular aggregates through both autophagy and the UPS.

In fALS and sALS, mutation of the sequestosome 1 / p62 protein encoded by *SQSTM1* has been identified. p62 is a multifunctional protein and is known to be particularly involved in degradation systems. p62 is an adapter protein that connects autophagosomes to substances that are selectively degraded. Therefore, it is thought that there is a problem in the degradation mechanism in ALS patients who have mutated p62 protein. In addition, p62 protein is known as a marker of autophagy and is degraded with the activation of autophagy. A previous study showed that the protein level of p62 decreases with increasing protein levels of LC3, which is an activation marker of autophagy in vivo and in vitro. In this study, p-CA upregulated LC3 and downregulated p62; therefore, p-CA induces autophagy.

Several studies have reported that the excessive oxidative stress is caused by the accumulation of misfolded SOD1 [12]. Then, oxidative stress results in a further increase of the formation of cytoplasmic aggregates of SOD1^mut^ [10,11]. This impaired protein quality control situation causes a vicious cycle in the motor neurons of ALS patients. Therefore, reducing excessive oxidative stress represents a potential therapeutic approach. Only riluzole and edaravone are widely known as approved drugs for ALS [52]. Edaravone is a scavenger of ROS, especially of the hydroxyl radicals [53]. From these results, both autophagy and antioxidative effects reduced the intracellular aggregation of SOD1^mut^ and prevented SOD1^mut^-associated neurotoxicity.

Many studies have showed serious mitochondrial disorder in SOD1^mut^-related ALS models [37,54,55]. Insoluble SOD1^mut^ aggregation is localized in mitochondria, especially in the outer membrane. Thus, SOD1^mut^ directly affects mitochondrial function [56,57]. The relation of ROS produced from mitochondria to mitochondrial oxidative injury is provided by this study. The ROS produced from mitochondria in SOD1^mut^-induced mitochondrial dysfunction cause a vicious cycle of injury.

In our results, p-CA degraded SOD1 aggregates and suppressed their formation via the activation of autophagy. In addition, several studies have reported that SOD1^mut^ aggregates accumulate in organelles such as mitochondria and the ER, causing excessive oxidative stress and ER stress and leading to neuronal cell death. Therefore, we suggest that p-CA has a neuroprotective effect against SOD1^mut^-induced neurotoxicity by suppressing oxidative stress and ER stress via autophagy.

In conclusion, to the best of our knowledge, this is the first study to investigate how p-CA prevents SOD1^mut^-associated neurotoxicity in ALS cell models. Although p-CA has many beneficial physiological activities, its effect on ALS pathogenesis—particularly SOD1 toxicity—was previously still unknown. This study shows that p-CA has neuroprotective effects against SOD1^mut^-associated neurotoxicity through the activation of autophagy. p-CA is an active component of Brazilian green propolis. Brazilian green propolis and its ingredient, p-CA, have the potential to delay the onset and/or the progression of ALS. p-CA will be very promising as a component of combination therapy for ALS in the future. This study provides an important contribution because the usefulness of p-CA against SOD1^mut^-induced neurotoxicity has been newly made clear.

## 4. Materials and Methods

### 4.1. Culture, Construct, and Transfection Cell Lines

We used the same procedures as those used in our previous study [23,36]. Briefly stated, expression plasmids (pmCherry-N1, Clontech Laboratories Inc (Mountain View, CA, USA) harboring variants of human SOD1 (wild-type (WT) or mutant (G85R)) were prepared as reported previously [23,36]. The mouse neuroblastoma Neuro2a (N2a) cell line was obtained from Public Health England (London, UK). For culturing, N2a culture cells were kept in a humidified atmosphere of 5% CO_2_ at 37 °C and maintained in Dulbecco’s modified Eagle medium (DMEM, Wako Pure Chemical Industries Ltd., Osaka, Japan) containing 10% (*v*/*v*) fetal bovine serum (FBS, Thermo Fisher Scientific Inc., Waltham, MA, USA). N2a cells were passaged by trypsinization every 3–4 days. The transfection of mCherry, SOD1^WT^–mCherry, and SOD1^G85R^–mCherry in N2a cells was performed using Lipofectamine 2000 according to the manufacturer’s instructions (Thermo Fisher Scientific Inc., Waltham, MA, USA).

### 4.2. Antibodies

For biochemical analysis, mouse monoclonal anti-CHOP (1:1000), rabbit polyclonal anti-LC-3 (1:1000), rabbit polyclonal anti-p62 (1:1000), and rabbit polyclonal anti-BiP (1:1000) were purchased from Cell Signaling Technology (Danvers, MA, USA). Mouse monoclonal anti-β-actin (1:2000) was purchased from Santa Cruz Biotechnology. Mouse monoclonal anti-mCherry (1:2000) was purchased from Clontech Laboratories Inc.

### 4.3. Thiazolyl Blue Tetrazolium Bromide (MTT) Assay and Lactate Dehydrogenase (LDH) Release Assay

We used the same procedures as those used in our previous study [23,36]. In short, the N2a cells were transfected with each plasmid. After 24 h, for differentiation, the culture medium was replaced for 48 h with differentiation medium [23,36] and incubated with or without p-CA (1 nM, 100 nM, or 1 µM). Cell viability was measured using a Cell Counting Kit-8, following the protocol (Wako Pure Chemical Industries Ltd., Osaka, Japan). After culturing N2a cells in differentiation medium for 48 h with or without p-CA, the amount of LDH was measured in the culture supernatant. Cell toxicity was measured using an LDH assay kit, following the protocol (Wako Pure Chemical Industries Ltd., Osaka, Japan).

### 4.4. Measurement of the Aggregation Rate

As stated above, N2a cells were transfected with each plasmid. After 24 h, the cells were treated with or without p-CA (1 nM, 100 nM, or 1 µM) for 24 h. After fixation with 4% paraformaldehyde, subcellular aggregation images were acquired using an IN Cell Analyzer 2200 high-content imaging cytometer (GE Healthcare, Buckinghamshire, UK). For measuring the number of aggregates, we used IN Cell Investigator (GE Healthcare, Buckinghamshire, UK). In each experiment, at least 3000 cells were counted.

### 4.5. Biochemical Analysis of Cell Culture Lysates

Regarding protein extraction from cells and Western blot protocol, we used the same procedures as those used in our previous study [23,36]. Briefly, the cells were lysed with TNE lysis buffer (50 mM Tris-HCl (pH. 7.4), 150 mM NaCl, 1 mM ethylenediaminetetraacetic acid, protease inhibitor cocktail) containing 1% Triton X-100 and were then centrifuged at 15,000 *g* for 5 min at 4 °C. The lysate supernatant was defined as the Triton-soluble fraction. Following centrifugation, the remaining deposition was resuspended with TNE lysis buffer containing 2% sodium dodecyl sulfate (SDS) (and defined as the Triton-insoluble fraction) [23,36,47]. Cell lysates were resolved by SDS-PAGE and transferred to a PVDF membrane. The proteins were detected by using an ECL system (GE Healthcare, Buckinghamshire, UK). The chemiluminescence was detected using LAS3000 mini films (Fuji film, Tokyo, Japan). ImageJ software (version 1.48, NIH, New York, NY, USA) was used to measure the band density.

### 4.6. Reactive Oxygen Species (ROS) Production

We performed the CellROX and MitoSOX assay based on previous studies [23,36]. Briefly stated, to detect SOD1^mut^-induced reactive ROS production, we used CellROX^®^ Green (Thermo Fisher Scientific Inc., Waltham, MA, USA), a fluorogenic probe designed to reliably measure ROS, and MitoSOX^®^ Red (Thermo Fisher Scientific Inc., Waltham, MA, USA), a fluorogenic probe designed to reliably measure ROS, especially superoxide anion in mitochondria, according to the protocols.

### 4.7. ESR Analysis

We performed ESR analysis based on previous studies [23,36]. Briefly stated, to detect the typical spectra of DMPO-OH spin, we generated the hydroxyl radicals via the Fenton reaction. The hydroxyl radicals was generated by mixing 72 mM DMPO (50 μL), 2 mM H_2_O_2_ (50 μL), and 0.2 mM FeSO_4_ (50 μL). The solution was transferred into an ESR spectrometry cell.

### 4.8. Statistical Analysis

The experimental results were analyzed by one-way ANOVA and Bonferroni/Dunn testing (StatView, Abacus, Baltimore, MD, USA). In this study, *p* < 0.05 was considered significant.

## Figures and Tables

**Figure 1 ijms-20-02942-f001:**
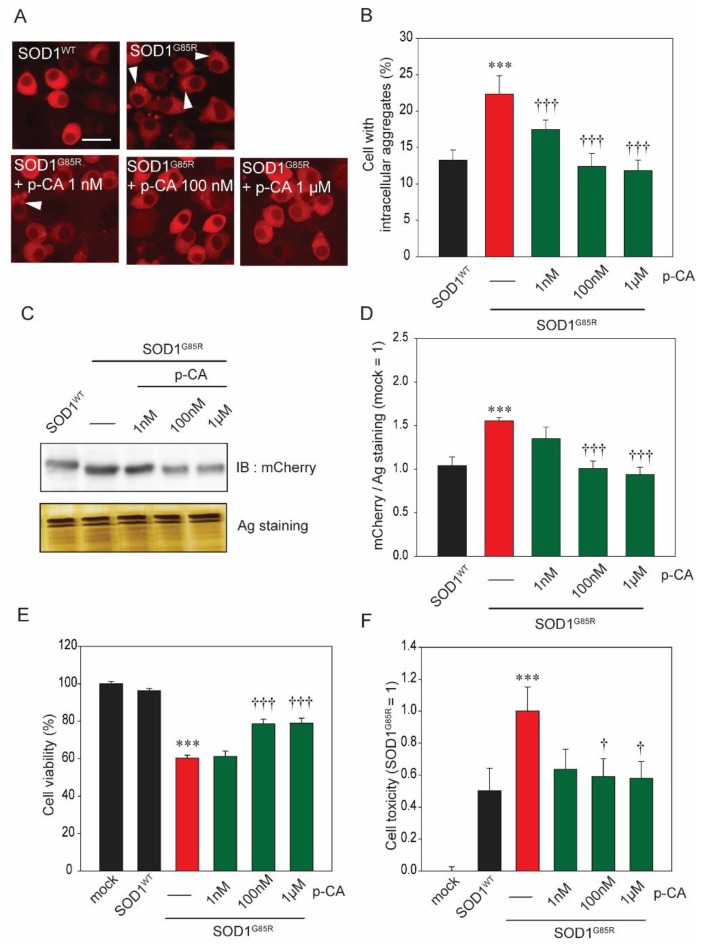
p-Coumaric acid (p-CA) reduced subcellular aggregation of mutant copper–zinc superoxide dismutase 1 (SOD1^mut^) and protected against SOD1^mut^-induced neurotoxicity. (**A**) Imaging of cytoplasmic mCherry–SOD1 aggregates (white arrowheads) in the presence or absence of p-CA in N2a cells. (**B**) Quantified analysis of imaging. (**C**) Immunoblot analysis of mCherry with a 1% Triton X-100 insoluble fraction. (D) Densitometric quantification of mCherry. (**E**) Cell viability was measured by MTT assay. (**F**) Cell toxicity was measured by LDH assay. Differences were evaluated by one-way ANOVA (mean ± SEM, *n* = 3). *** *p* < 0.001 vs. SOD1^WT^, ^†††^
*p* < 0.001, ^†^
*p* < 0.05 vs. SOD1^G85R.^ p-CA: p-coumaric acid. Scale bar: 10 µm.

**Figure 2 ijms-20-02942-f002:**
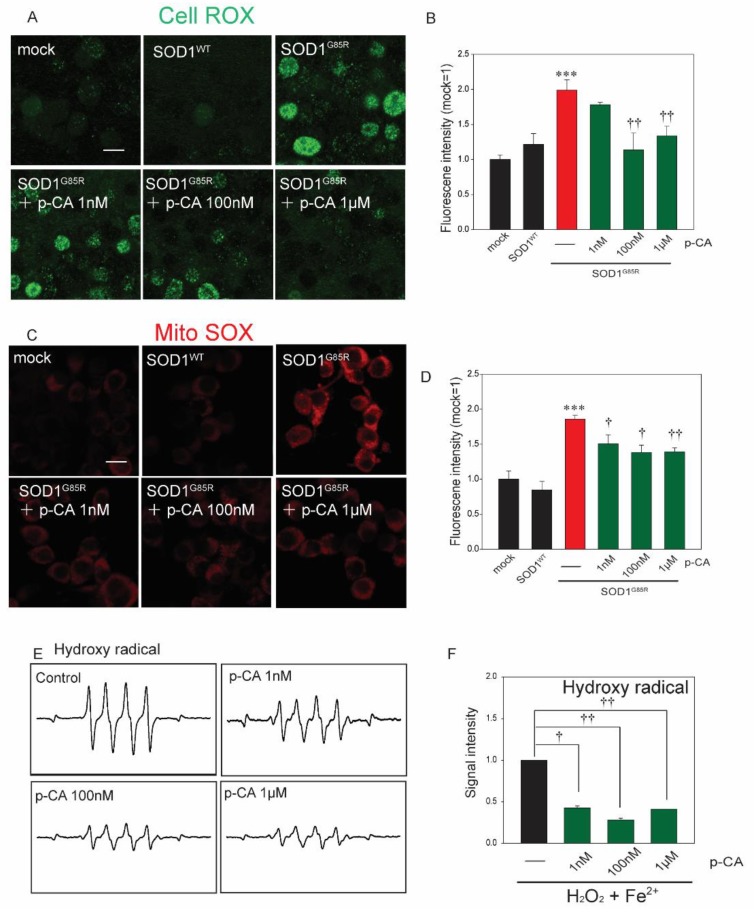
p-CA reduced mutant-SOD1-related oxidative stress. (**A**) Confocal imaging of CellROX in N2a cells transfected with mCherry–SOD1^G85R^ with p-CA for 24 h. (**B**) Quantified analysis of CellROX using Image J. (**C**) Confocal imaging of MitoSOX in N2a cells transfected with mCherry–SOD1^G85R^ with p-CA for 24 h. (**D**) Quantified analysis of MitoSOX using Image J. (**E**) Typical spectra of DMPO-OH spin generated from H_2_O_2_ plus Fe^2+^ in the absence (control) or presence of p-CA. (**F**) The amount of hydroxy radicals was semi-quantitatively measured as the formation of DMPO-OH spin adducts by ESR spectrometry. Differences were evaluated by one-way ANOVA (mean ± SEM, *n* = 3). *** *p* < 0.001 vs. SOD1^WT^, ^††^
*p* < 0.01, ^†^
*p* < 0.05 vs. SOD1^G85R^. Scale bar: 10 µm. p-CA: p-coumaric acid.

**Figure 3 ijms-20-02942-f003:**
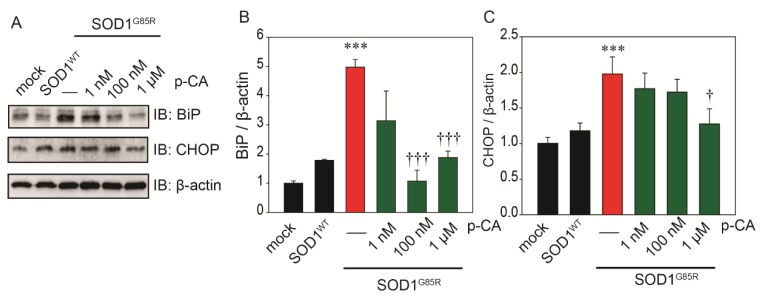
p-CA reduced SOD1^mut^-related ER stress. (**A**) Immunoblot analysis of BiP and CHOP relating to ER stress. (**B**,**C**) Densitometric quantification of BiP (**B**) and CHOP (**C**). Differences were evaluated by one-way ANOVA (mean ± SEM, *n* = 3). *** *p* < 0.001 vs. SOD1^WT^, ^†††^
*p* < 0.001, ^†^
*p* < 0.05 vs. SOD1^G85R^. p-CA: p-coumaric acid.

**Figure 4 ijms-20-02942-f004:**
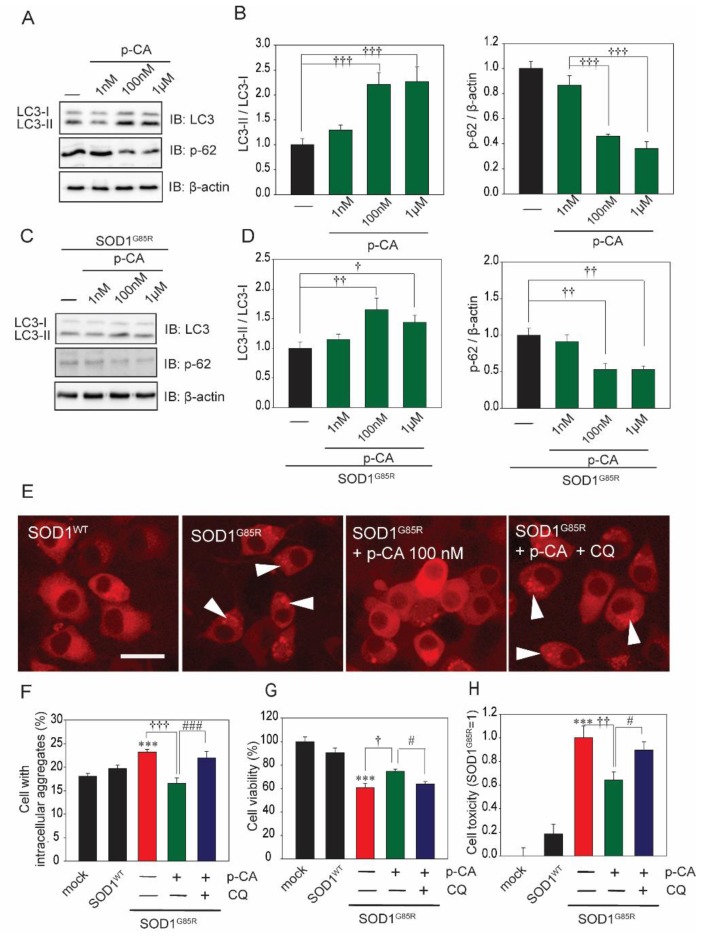
p-CA prevented SOD1^mut^-associated neurotoxicity through autophagy. (**A**) Immunoblot analysis of LC3 and p62 relating to autophagy. (**B**) Densitometric quantification of LC3 and p62. ^†††^
*p* < 0.001 vs. control. (**C**) Immunoblot analysis of LC3 and p62 relating to autophagy with SOD1^G85R^-expressing cells. (**D**) Densitometric quantification of LC3 and p62. ^†††^
*p* < 0.001 vs. control. (**E**) Imaging of cytoplasmic mCherry–SOD1 aggregates (white arrowheads) in the N2a cells with CQ (1 nM) before p-CA (100 nM) treatment. (**F**) Quantified analysis of imaging. (G) Cell viability was measured by MTT assay. (H) Cell toxicity was measured by LDH assay. Differences were evaluated by one-way ANOVA (mean ± SEM, *n* = 3). *** *p* < 0.001 vs. SOD1^WT^, ^†††^
*p* < 0.001, ^††^
*p* < 0.01, ^†^
*p* < 0.05. vs. SOD1^G85R^, ^###^
*p* < 0.001, ^#^
*p* < 0.05. vs. SOD1^G85R^ + p-CA. Scale bar: 10 µm. p-CA: p-coumaric acid, CQ: chloroquine.

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
