# Peer review of "p-Coumaric Acid Has Protective Effects against Mutant Copper–Zinc Superoxide Dismutase 1 via the Activation of Autophagy in N2a Cells"

_ijms, 2019, doi:10.3390/ijms20122942_

Reviewer 1 Report

 Based upon my previous comments the authors have done significant changes, although some of the issue were not resolved but still I am happy with the revision. I have no further questions.

Author Response

Reply to comments by Reviewer #1

We would appreciate the reviewer’s very useful comments and thoughtful critiques and questions.

We attach the final clear file.

Reviewer 2 Report

The manuscript is greatly improved, authors made an effort to address all reviewers' concerns. 

However, some edits to improve English are still needed.

Author Response

Reply to comments by Reviewer #2

Comment 1- The manuscript is greatly improved, authors made an effort to address all reviewers' concerns. However, some edits to improve English are still needed.

Reply 1- We appreciate reviewers’ interest in our work, the detailed evaluations and the helpful suggestions. According your comment, we got English proofreading and corrected our manuscript carefully. Those your comments were all valuable and very helpful for revising and improving our paper.

We attach the corrected file with the red color.

Reviewer 3 Report

The manuscript by Ueda et al. reports that p-coumaric acid in cell model provides neuroprotection in mutant copper-/zinc superoxide dismutase 1 via the activation of autophagy. Thus the manuscript contains a set of novel valuable information, is well written and easy to follow. I have only a couple of concerns:

- The authors performed their experiments in cell models only. In my opinion this fact should be pointed within the title.

- Page 2, Lane 32: ‘Triton-X insoluble SOD1’; page 3, lane 5: ‘Triton-100’. Please modify to Triton X-100. More generally, in my opinion, ‘Triton X-100 insoluble SOD1 aggregates’ should be defined and appropriate reference should be included.

Author Response

Reply to comments by Reviewer #3

Comment 1- The authors performed their experiments in cell models only. In my opinion this fact should be pointed within the title.

Reply 1- According to your comments, we change the our manuscript title (please see below).

“p-Coumaric acid has protective effects against mutant copper-zinc superoxide dismutase 1 via the activation of autophagy in N2a cells.”

Comment 2- Page 2, Lane 32: ‘Triton-X insoluble SOD1’; page 3, lane 5: ‘Triton-100’. Please modify to Triton X-100. More generally, in my opinion, ‘Triton X-100 insoluble SOD1 aggregates’ should be defined and appropriate reference should be included.

Reply 2- We agree with your comments. According to your comment, we modify the manuscript (page 2, lane 38; page 3, lane 5).

 We appreciate the reviewer for the thoughtful comment. According to your comment, we define Triton X-100 insoluble SOD1 aggregates’ and add appropriate some references (page 8, lane 41- page 9, lane 2).

We attach the corrected file with red color.

This manuscript is a resubmission of an earlier submission. The following is a list of the peer review reports and author responses from that submission.

Round  1

Reviewer 1 Report

The study entitled - p-Coumaric acid provides neuroprotection in copper-zinc superoxide dismutase 1 associated with amyotrophic lateral sclerosis model cell via the activation of autophagy- by Ueda et al., is an interesting study however poorly presented. Below you will find my suggestions to improve the overall quality of the MS.

Major comments-

a- Introduction and abstracts need major corrections for example

1-The title need to be changed

2- page 30-31-The mutations in copper-zinc superoxide dismutase 1 (SOD1) are the major autosomal dominant inherited cause for ALS-  Mutation in c9orf 72 constitute for 30% of fALS and SOD1- 20%, please refer recent reviews on genetics of ALS.

3- the link between oxidative stress, ER stress and autophagy is not well discribed before asking the questions what the authors are addressing in this MS. The introduction needs to be throughly revised with specific questions.

Results:

1- In fig.1;The interpretation that p-CA reduced cytoplasmic aggregates of SOD1mut, is solely based on images captured/ analysed by Cell Analyzer 2200. Its not clear how they programmed the instrument to detect the aggregates Vs soluble. Biochemical evidence, such as gel top aggregation, filter trap methods is obsolutly nesessary to further confirm the facts.

2- In fig 2-3, for figure 2 the authors need to further confirm the findings with westernblot analysis whether the oxidative stress is minimized by the drug. Similarly BIP immunoblots are not sufficient to confirm the reduction of ER stress, other markers should be included. CHOP reduction is not very informative. The figure will be enhanced if they can include immunofluorescence studies probably using Cell Analyzer 2200 after staining with ER stress markers or so.

3- Fig 4  Colabelling with p62, LC3 should be used together with SOD1mut

Disscussion needs more focus SPECIALLY on the relationship between ER stress, proteotoxic stress and autophagy and the effect of the drug used.

Author Response

We appreciate reviewers’ interest in our work, the detailed evaluations and the helpful suggestions. Those comments are all valuable and very helpful for revising and improving our paper. We hereby address the reviewers’ concerns as below and made substantial changes in the manuscript.

Major comments-
a- Introduction and abstracts need major corrections for example
Comment 1-The title need to be changed
Reply 1- According to your and another reviewer’s comments, we simplify the title (please see below).
“p-Coumaric acid provides neuroprotection in mutant copper-zinc superoxide dismutase 1 via the activation of autophagy”

Comment 2- page 30-31-The mutations in copper-zinc superoxide dismutase 1 (SOD1) are the major autosomal dominant inherited cause for ALS- Mutation in c9orf 72 constitute for 30% of fALS and SOD1- 20%, please refer recent reviews on genetics of ALS.
Reply 2- According your comment, we confirm the recent review on genetics of ALS (Mathis et al., 2019). fALS is identified by mutations in several genes, including SOD1, C9ORF72, TARDBP, and FUS. In Europe, the most common fALS mutations are C9ORF72 (33.7%), SOD1 (14.8%), TARDBP (4.2%), and FUS (2.8%). In contrast, the most common Asian fALS mutations are SOD1 (30%), FUS (6.4%), C9ORF72 (2.3%), and TARDBP (1.5%). We improve the introduction part of the revised manuscript (page 1. Lane 30-33).

Comment 3- the link between oxidative stress, ER stress and autophagy is not well discribed before asking the questions what the authors are addressing in this MS. The introduction needs to be throughly revised with specific questions.

Reply 3-
SOD1mut aggregates accumulated in organelle such as mitochondria and ER (Shi et al., 2010; Kikuchi et al., 2006). Accumulation of SOD1mut in these organelle induced oxidative stress and ER stress, which then cause further increases in the insoluble SOD1mut aggregate formation (D’Amico et al., 2013; Rakhit et al., 2002). Several studies have shown that excessive oxidative stress leading to neuronal cell death is caused by accumulation of misfolded SOD1 (Vehviläinen et al., 2014; Nishitoh et al., 2008). In addition, activation of autophagy suppresses motor neuron cell death via clearance of the SOD1mut aggregations in cellular and mouse models of ALS (Wong et al., 2010; Chhangani et al., 2016). Therefore, activation of autophagy may represent a therapeutic approach for ALS. We add the introduction part in the revised manuscript (page 1. lane 42-page 2. lane 6).

Results:
Comment 1- In fig.1;The interpretation that p-CA reduced cytoplasmic aggregates of SOD1mut, is solely based on images captured/ analysed by Cell Analyzer 2200. Its not clear how they programmed the instrument to detect the aggregates Vs soluble. Biochemical evidence, such as gel top aggregation, filter trap methods is obsolutly nesessary to further confirm the facts.
Reply 1- According to your comment, we examined whether p-CA reduced cytoplasmic insoluble aggregates of SOD1mut by western blot which is a biochemical method. We used the same procedures as those used in the previous our study (Ueda et al., 2018; Ueda et al., 2017). Briefly, after 24 h of transfection to N2a cells each vector, the cells were treated with or without p-CA. The cells were lysed with TNE lysis buffer containing 1% Triton-X. After centrifuge, the remaining deposition was resuspended with TNE lysis buffer containing 2% sodium dodecyl sulfate (SDS) (defined as Triton-insoluble fraction). Cell lysates were resolved by SDS-PAGE and transferred to PVDF membrane. The proteins were detected by using ECL system. As a result, we found that p-CA reduced insoluble SOD1mut aggregates. We add the results of figure 1C, D in the revised manuscript (page 2, lane 33, 34).

Comment 2- In fig 2-3, for figure 2 the authors need to further confirm the findings with westernblot analysis whether the oxidative stress is minimized by the drug. Similarly BIP immunoblots are not sufficient to confirm the reduction of ER stress, other markers should be included. CHOP reduction is not very informative. The figure will be enhanced if they can include immunofluorescence studies probably using Cell Analyzer 2200 after staining with ER stress markers or so.

Reply 2- We appreciate the reviewer for the thoughtful comments about the crucial problem. We examined whether p-CA reduced SOD1-induced oxidative stress using cellROX and mitoSOX probes in the previous MS. As an alternative to Western blot, we also confirmed whether p-CA directly scavenged ROS with ESR assay. We found that p-CA significantly attenuated the signal intensity of the hydroxy radical. We add the results as figure 2E, F in the revised manuscript (page 3, lane 16-18)
In figure 3, we performed neurochemical analysis using the BIP and CHOP antibody which is the most famous ER stress markers in the previous MS. We agree with your proposal, but unfortunately it takes a lot of time to consider the conditions for measuring ER stress with In cell Analyzer 2200. We would like to make it a future study.

Comment 3- Fig 4 Colabelling with p62, LC3 should be used together with SOD1mut.
Disscussion needs more focus SPECIALLY on the relationship between ER stress, proteotoxic stress and autophagy and the effect of the drug used.

Reply 3- We agree with your comments. According to your comment, we performed western blot using p62, LC3 antibody with N2a cells which expressing SOD1mut with or without p-CA. We add the results in figure 4C, D (page 5, lane 13-15).
We appreciate the reviewer for the thoughtful comment. As shown by our results, p-CA degraded SOD1 aggregates and suppressed their formation via activation of autophagy. Similarly, several studies have reported that SOD1mut aggregates accumulate in organelles such as mitochondria and ER, where they cause excessive oxidative and ER stress, leading to neuronal cell death. Therefore, we suggest that p-CA has neuroprotective effects against SOD1mut-induced neurotoxicity by suppressing oxidative and ER stress via autophagy. We add the discussion of this point in the revised manuscript (page 7, lane 44-48).

Reviewer 2 Report

The authors investigated the neuroprotective effects of p-Coumaric acid (p-CA) in N2a cells transfected with WT or SODG85R plasmids. They found that p-CA reduced the accumulation of SOD aggregates, promoted autophagy and increased cell viability. Although these results may be interesting, the authors repeated the study design they used to investigate other active ingredients of Brazilian green propolis (ref # 17). Both studies found very similar findings. Authors should comment on these similarities and discuss potential differences in biological pathways activated/inhibited by these compounds.

There are several concerns and/or questions:
1. The manuscript is written quite carelessly. There are incomplete sentences, missing commas, periods and capital letters at the beginning of the sentence to name just few issues.

2. Discussion is not adequate, especially around p62 and its role in autophagy and ALS.

3. Method section is also lacking important details; better description of cell cultures and transfections is needed. LDH assay also needs more details – was the LDH measured in cell culture media? Interestingly, there was no difference in cell viability between mock and SOD1WT groups (Fig 1C) but it seems there was a significant increase in LDH in SOD1WT (Fig 1D). Can author comment on this?

3. Fig 2A, C – pictures are of low quality. In addition, inserts with higher magnifications will be quite helpful.

Author Response

We would appreciate the Reviewer’s very useful comments and thoughtful critiques and questions. We revised the manuscript based on your comments.

The authors investigated the neuroprotective effects of p-Coumaric acid (p-CA) in N2a cells transfected with WT or SOD1G85R plasmids. They found that p-CA reduced the accumulation of SOD1 aggregates, promoted autophagy and increased cell viability. Although these results may be interesting, the authors repeated the study design they used to investigate other active ingredients of Brazilian green propolis (ref # 17). Both studies found very similar findings. Authors should comment on these similarities and discuss potential differences in biological pathways activated/inhibited by these compounds.
In our study, we revealed that p-CA induced autophagy. We have previously shown that kaempferol induced autophagy via AMPK-mTOR pathway and had a protective effect on SOD1mut-related neurotoxicity. Other reports have shown that p-CA activates AMPK (Kang et al., 2013; Yoon et al., 2013). Based on these reports, p-CA may induce autophagy via the AMPK–mTOR pathway. In addition, we previously found that kaempferol did not activate autophagy via the AKT–mTOR pathway (Ueda et al., 2017). The AKT signal induced mTOR activity and then inhibited autophagy (Li et al., 2016). Unfortunately, in our study, we did not examine whether p-CA inhibited the AKT signal. However, p-CA is known to inhibit the AKT signal (Kong et al., 2012). Thus, p-CA may induce autophagy via the AMPK–mTOR pathway and the AKT–mTOR pathway. We add the discussion of this point in the revised manuscript (page 7, lane 6-12).
There are several concerns and/or questions:

Comment 1- The manuscript is written quite carelessly. There are incomplete sentences, missing commas, periods and capital letters at the beginning of the sentence to name just few issues.
Reply 1- Thank you for your kind comment, we correct the manuscript.

Comment 2- Discussion is not adequate, especially around p62 and its role in autophagy and ALS.

Reply 2- In fALS and sALS, mutation of the sequestosome 1/p62 protein encoded by SQSTM1 has been identified. p62 is a multifunctional protein known to be particularly involved in degradation systems. Further, it is an adapter protein that connects autophagosomes to substances that are selectively degraded. Therefore, there is thought to be a problem in the degradation mechanism in ALS patients with mutated p62 protein. In addition, p62 is a marker of autophagy and is degraded with the activation of autophagy. In this our study, p-CA upregulated LC3 and downregulated p62, indicating that p-CA induces autophagy. We add the discussion of this point in the revised manuscript (page 7, lane 23-31).

Comment 3- Method section is also lacking important details; better description of cell cultures and transfections is needed. LDH assay also needs more details – was the LDH measured in cell culture media? Interestingly, there was no difference in cell viability between mock and SOD1WT groups (Fig 1C) but it seems there was a significant increase in LDH in SOD1WT (Fig 1D). Can author comment on this?

Reply 3- Thank you for your kind comment, we corrected the manuscript. We add the detail about cell culture, transfection and LDH assay (page 8, lane 10-14, 21, 27, 28).
We appreciate the Reviewer for the thoughtful questions about the crucial problem. In our study, we examined SOD1mut-iduced neurotoxicity with MTT assay and LDH assay. There was no difference in cell viability between mock and SOD1WT groups but it seems there was a significant increase in LDH in SOD1WT. As this reason, it can be considered that the measurement principle is different. MTT assay is metabolic activity as an indicator while LDH assay is cell membrane damage as an indicator.

Comment 4. Fig 2A, C – pictures are of low quality. In addition, inserts with higher magnifications will be quite helpful.
Reply 4- According to your comment, we insert the images with higher magnifications. We improve the figure 2A,C in the revised manuscript.

Reviewer 3 Report

The manuscript by Ueda et al. reports that p-Coumaric acid (the component from Brazilian green propolis extract) functions as neuroprotector in amyotrophic lateral sclerosis model via the activation of autophagy. The manuscript represents an elegant study, which is novel and has a value. I have only few recommendations:

- In my opinion the title of the article could be simplified;

- Abstract (lane 20) needs correction;

- The list of abbreviations is incomplete.

Author Response

We thank you for your interest in this paper. We improved our manuscript with reference to your very helpful comments. We hereby address the reviewers’ concerns as below and made substantial changes in the manuscript.

Comment 1- In my opinion the title of the article could be simplified;
Reply 1- We agree with your comments. According to your comment, we change the title (please see below).
“p-Coumaric acid provides neuroprotection in mutant copper-zinc superoxide dismutase 1 via the activation of autophagy”

Comment 2- Abstract (lane 20) needs correction;
Reply 2- We appreciate the reviewer for the thoughtful comments. According your comment, we modify the abstract (page1, lane 19, 20).

Comment 3- The list of abbreviations is incomplete.
Reply 3- Thank you for your kind comment, we correct the list of abbreviations (page 9, lane 21).